# *Ss*Cak1 Regulates Growth and Pathogenicity in *Sclerotinia sclerotiorum*

**DOI:** 10.3390/ijms241612610

**Published:** 2023-08-09

**Authors:** Lei Qin, Jieying Nong, Kan Cui, Xianyu Tang, Xin Gong, Yunong Xia, Yan Xu, Yilan Qiu, Xin Li, Shitou Xia

**Affiliations:** 1Hunan Provincial Key Laboratory of Phytohormones and Growth Development, Hunan Agricultural University, Changsha 410128, China; leiqin2020@stu.hunau.edu.cn (L.Q.); njy@stu.hunau.edu.cn (J.N.); chuxuantingnasha@stu.hunau.edu.cn (X.T.); xingong@stu.edu.cn (X.G.); yun0623@stu.edu.cn (Y.X.); 2Institute of Plant Protection, Hunan Academy of Agricultural Sciences, Changsha 410125, China; ck0601@stu.hunau.edu.cn; 3Michael Smith Laboratories, University of British Columbia, Vancouver, BC V6T 1Z4, Canada; yan.xu@msl.ubc.ca; 4Department of Botany, University of British Columbia, Vancouver, BC V6T 1Z4, Canada; 5Department of Life Science, Hunan Normal University, Changsha 410081, China; qiu730822@163.com

**Keywords:** *Sclerotinia sclerotiorum*, *Ss*Cak1, sclerotia, appressoria, pathogenesis, HIGS

## Abstract

*Sclerotinia sclerotiorum* is a devastating fungal pathogen that causes severe crop losses worldwide. It is of vital importance to understand its pathogenic mechanism for disease control. Through a forward genetic screen combined with next-generation sequencing, a putative protein kinase, *Ss*Cak1, was found to be involved in the growth and pathogenicity of *S. sclerotiorum*. Knockout and complementation experiments confirmed that deletions in *SsCak1* caused defects in mycelium and sclerotia development, as well as appressoria formation and host penetration, leading to complete loss of virulence. These findings suggest that *Ss*Cak1 is essential for the growth, development, and pathogenicity of *S. sclerotiorum*. Therefore, *Ss*Cak1 could serve as a potential target for the control of *S. sclerotiorum* infection through host-induced gene silencing (HIGS), which could increase crop resistance to the pathogen.

## 1. Introduction

*Sclerotinia sclerotiorum* is a phytopathogenic fungus which has a wide host range and can infect more than 600 species of plants, including rape, potato, cotton, tomato, soybean, and other important crops [1,2,3]. *S. sclerotiorum* causes stem rot, resulting in the death of host tissues. Its virulence involves the release of toxins (such as oxalic acid (OA) and cell wall-degrading enzymes (CWDEs) for infection initiation, followed by the extraction of nutrients from host cells [4]. As a necrotrophic pathogen, *S. sclerotiorum*, however, has a brief biotrophic phase that begins approximately 12–24 h after infection [3,5]. During this stage, *S. sclerotiorum* establishes compatibility with the host by suppressing or disrupting its defense barriers [6]. Subsequently, the subcutaneous hyphae of *S. sclerotiorum* spread to multiple cell layers. With successful colonization by branched hyphae, *S. sclerotiorum* enters a necrotrophic phase, producing large amounts of reactive oxygen species, toxins, and CWDEs, leading to the development of host cell death and necrotic symptoms [5,7,8].

The complex appressorium is a key infection structure for establishing infection. It is formed by hyphae that undergo swelling, slow growth, and continuous branching. Previous studies have shown that these cells can enhance the pathogen’s adhesion to the host surface [9] and help penetrate the host epidermal barrier through mechanical pressure or enzymatic degradation [10]. The hyphae forming complex appressoria often become flattened and increase in diameter, from which narrow penetration pegs are formed to complete penetration [9,11]. Subsequently, subcutaneous infection hyphae are produced and differentiated. These hyphae grow horizontally beneath the epidermis, forming the colonization leading edge [3].

To date, host-induced gene silencing (HIGS) has become a promising way to control fungal diseases, including sclerotinia stem rot. Andrade et al. [12] demonstrated for the first time that the HIGS-mediated chitin synthase gene (*CHS*) enhanced T1 generation resistance to *S. sclerotiorum* in tobacco. In subsequent independent studies, by selecting *Sscnd1*, *Ssoah1*, *ABHYRDOLASE-3* and *SsTrxR1* as the target genes of HIGS, the resistance of plants to *S. sclerotiorum* was successfully improved [6,13,14,15]. Therefore, conducting in-depth research on the pathogenic mechanisms of *S. sclerotiorum* and screening of key genes, along with enhancing the host’s resistance to the pathogen through cross-kingdom RNA silencing, can potentially lead to effective control of sclerotinia stem rot (SSR).

Recently, Xu et al. developed a method combining forward genetic screening with high-throughput next-generation sequencing for the rapid discovery of new genes involved in sclerotia development. Some of these genes can be used as HIGS targets for disease control [16,17]. Here, it was performed as a genetic screen for virulence-related genes in a mutagenized population of *S. sclerotiorum* and to isolate a mutant strain with defects in growth, sclerotia development and pathogenicity. Then a candidate gene, *SsCak1*, was identified by NGS. Knockout and complementation experiments confirmed its involvement in the development of *S. sclerotiorum* mycelium, sclerotia, complex appressoria formation, and pathogenicity, making it a potential target for HIGS to enhance crop resistance against *S. sclerotiorum*.

## 2. Results

### 2.1. Identification of a Pathogenicity-Attenuated Mutant in S. sclerotiorum through Forward Genetic Screening

To study the pathogenic mechanism of *S. sclerotiorum* in depth, we carried out a forward genetics screen on a UV-mutagenized population of *S. sclerotiorum* [18]. Using leaves of lettuce (Appendix A), we aimed to identify mutants with virulence defects. Here we report on a mutant strain M14-9. On tobacco leaves, wild-type (WT) *S. sclerotiorum* caused severe leaf maceration 48 h after inoculation, whereas M14-9 did not (Figure 1A,B). Moreover, the M14-9 mutant exhibited markedly impaired growth (Figure 1C). The growth rate of M14-9 was much lower than that of WT (Figure 1D), and a large number of aerial hyphae were produced at the early stage of colony formation. In addition, the hyphae of the M14-9 appeared dense and short under the microscope (Figure 1E). M14-9 is deficient in sclerotia formation, with fewer sclerotia numbers per dish and lower individual sclerotia weight than WT (Figure 1F,G). Thus, M14-9 shows dual defects in pathogenicity and growth.

### 2.2. Sscle_11g085070 Is the Candidate Mutated Gene of M14-9

To determine the causal mutation responsible for the phenotype of M14-9, the whole genome of M14-9 was re-sequenced and analyzed by NGS. Gene sequences of two other mutants, 49-23 and A14-11, which were identified in the same screen and sequenced, were used as negative controls to exclude background mutations in M14-9. The NGS data were then analyzed with a modified NGS sequence analysis pipeline in *S. sclerotiorum* [19]. After removing background mutations, synonymous mutations, intron mutations, and intergenic mutations, there were three remaining SNP mutations that became the main candidate mutations for subsequent analysis (Appendix A).

To further narrow down the candidates, protein sequences encoded by these genes were analyzed to examine the functions of their orthologs in closely related species. Literature research revealed that *sscle_11g085070*, whose ortholog, *Fg*Cak1, from *Fusarium graminearum*, is involved in hyphal development and pathogenicity [20]. Therefore, *SsCak1* was initially tested in *S. sclerotiorum*. Sanger sequencing of *SsCak1* using DNA from M14-9 confirmed a single nucleotide mutation in the second exon of the gene, resulting in a premature stop codon, consistent with the NGS sequencing results (Appendix A).

Furthermore, phylogenetic analysis results revealed that the homologous proteins of *Ss*Cak1 (protein accession number, APA13737.1/XP_001590945.1) are relatively conserved in fungi, and widely present in *Botrytis cinerea* and other pathogenic fungi (Figure 2A). All Cak1s have two conserved eukaryotic protein kinase domains, VIB and VIII (Figure 2B). According to the traditional classification method, *Ss*Cak1 belongs to the CMGG group of eukaryotic protein kinases. To gain a preliminary insight into the role of *Ss*Cak1 in fungal development, RT-qPCR analysis was conducted to determine the abundance of *SsCak1* mRNA in different growth stages of *S. sclerotiorum*. As shown in Figure 2C, *SsCak1* was expressed constitutively in different developmental stages. However, the highest expression was observed in the mature sclerotium stage (7 dpi) (Figure 2C). When inoculated onto leaves of *N. benthamiana*, the expression of *SsCak1* was significantly upregulated from nine hpi to twenty-four hpi (Figure 2D), indicating that *SsCak1* was strongly induced during *S. sclerotiorum* infection. These results suggest that *SsCak1* may play an important role in the growth, development, and pathogenicity of *S. sclerotiorum*, and the mutated phenotype of M14-9 may be caused by the truncation of *SsCak1*.

### 2.3. SsCak1 Deletion Impairs Mycelial Growth and Sclerotia Development

To test the gene function of *SsCak1*, we generated a knockout mutant *Sscak1* in the background of the WT strain by homologous recombination, and obtained two independent complementation strains, *Sscak1-C1* and *Sscak1-C5*, by introducing a WT copy of *SsCak1* into the knockout mutant (Appendix A and Figure 3A). PCR and RT-qPCR showed that *SsCak1* was completely deleted in the knockout mutant, and its transcript was absent, while the two complementation strains restored the transcript levels (Appendix A and Figure 3D). The mycelial morphology of *Sscak1* was largely similar to that of M14-9, showing dense mycelial branching and slower growth rate (Figure 3B,E), while the complementation strains *Sscak1-C1* and *Sscak1-C5* regained the WT phenotypes. After 15 days of growth on PDA medium, the *SsCak1* knockout mutant exhibited abnormal sclerotia development on the colony surface (Figure 3C). Although *Sscak1* successfully formed sclerotia, the number of sclerotia produced per plate was significantly reduced (Appendix A), and the average mass of each sclerotium was also lighter than that of WT (Appendix A), indicating that loss of *SsCak1* impairs growth of mycelium and sclerotia development in *S. sclerotiorum*.

### 2.4. Knockout of SsCak1 Results in Complete Loss of Pathogenicity in S. sclerotiorum

For pathogenicity tests, the WT, mutant M14-9, *Sscak1*, and complementation strains *Sscak1-C1* and *Sscak1-C5* were inoculated onto intact and wounded leaves of *A. thaliana* (Figure 4A) and *N. benthamiana* (Figure 4B). After 48 h, on intact leaves, neither M14-9 nor *Sscak1* was able to infect, whereas the complementation strains caused large areas of infection similar to that of WT (Figure 4C,D). On wounded leaves, M14-9 and *Sscak1* caused mild infection damage compared to the WT and complementation strains (Figure 4C,D). These results confirmed that *SsCak1* is essential for the pathogenicity of *S. sclerotiorum*, which is likely to involve both pre- and post-penetration regulations.

### 2.5. SsCak1 Is Essential for Appressorium Formation and Penetration

To further investigate the details of the pathogenicity defects in *Sscak1* mutants, their oxalate production, appressoria development, and penetration ability were examined. First, fresh mycelium blocks were inoculated on the PDA medium containing bromophenol blue as a pH indicator. After 24 h, all areas of the mycelium inoculated, including WT, M14-9, *Sscak1*, *Sscak1-C1*, and *Sscak1-C5*, had turned from blue to yellow (Appendix A), suggesting that *Ss*Cak1 does not affect oxalate production. However, M14-9 and *Sscak1* failed to produce appressorium on glass slides, while the WT strain formed mature appressorium cell structures upon contact with the slide (Figure 5A,B). Onion epidermal penetration assays further revealed that the WT strain formed numerous appressoria and invasive hyphae, whereas M14-9 and *Sscak1* failed to produce them, and the hyphae were unable to penetrate the onion epidermis (Figure 5C). Therefore, *SsCak1* is an important factor in appressoria development, and the defect in appressorium development and permeability is likely to be the main reason for the loss of pathogenicity in *Sscak1*.

### 2.6. HIGS of SsCak1 in N. benthamiana Enhances Resistance to S. sclerotiorum

Conserved and functionally important genes can sometimes be used as target genes for the control of pathogenic microorganisms using HIGS. Since sequence alignment and phylogenetic tree analysis showed that there was no gene with a similar sequence to *SsCak1* in plants, *SsCak1* may be a potential target for stem rot control using HIGS. Thus, tobacco rattle virus (TRV)-mediated transient silencing of *SsCak1* was performed in *N. benthamiana*. Here, three *Agrobacterium* constructs, including a positive control construct *pTRV2:PDS*, a negative control construct *pTRV2:GFP* and an experimental group *pTRV2:SsCak1* construct, were agro-infiltrated into *N. benthamiana* leaves. After seven days, the top leaves of *N. benthamiana* showed chlorosis with *pTRV2:PDS* infiltration, which gradually expanded to the entire plant (Figure 6A). At the same time, the expression of the target genes can be quantified by semi-quantitative PCR (Figure 6B), demonstrating the feasibility of the gene silencing system. After 14 days of TRV treatment, plants were inoculated with the WT strain of *S. sclerotiorum* (Figure 6C). Compared with that in the control leaves (*TRV:GFP*), the lesion area on *TRV:SsCak1* was reduced by 62% at 24 hpi (Figure 6D). Meanwhile, expression of *SsCak1* in *TRV:SsCak1*-treated leaves was also reduced to 60% of control leaves (*TRV:GFP*) (Figure 6E). These results suggest that silencing of *SsCak1* by TRV–HIGS does reduce virulence and enhances host resistance against *S. sclerotiorum*.

## 3. Discussion

*S. sclerotiorum* has garnered significant attention from researchers due to its substantial impact on *Brassica napus* and other economically important crops [3,21]. However, prior investigations into the growth, development, and virulence of *S. sclerotiorum* have typically relied on homologous proteins from extensively studied pathogens to unravel pathogenicity-associated elements within the organism. Consequently, this approach has inherent limitations, impeding the pace of research progress on *S. sclerotiorum*. In this study, successful identification of a putative CDK-activating kinase was achieved by employing a combination of forward genetic screening technology and NGS sequencing. And found that it is necessary for the growth and virulence of *S. sclerotiorum*.

Reversible phosphorylation is a major mechanism by which environmental and biochemical stimuli affect protein function and gene expression. Most eukaryotic protein kinases (ePKs) phosphorylate serine, threonine, or tyrosine and have a highly conserved catalytic domain consisting of 12 subdomains that make up the ATP-binding lobe (subdomains I–V) and peptide binding and phosphotransfer lobes (subdomains VI–XI). Subdomains VIB and VIII are involved in peptide substrate recognition, and conserved amino acids within them are used to classify ePKs into functional groups [22,23,24]. According to the multiple sequence alignment of homologues, Cak1 is relatively conserved in fungi and has a conserved eukaryotic kinase domain, which can be classified into CMGG groups according to their subdomains VIB and VIII. *S. sclerotiorum* ePKs CMGG group is widely produced during mycelium development, virulence formation, sclerotia development and host penetration. SMK1, another ePKs CMGG group by RNA-silencing, showed impaired sclerotia formation in *S. sclerotiorum*, while in other pathogens, inactivation of the *SMK1* homologous gene resulted in loss of pathogenicity due to the inability to form appressorium [25]. Disruption of the SMK3, an important ePKs of the CMGG group in *S. sclerotiorum*, results in an inability to aggregate and form appressorium leading to a severe reduction in virulence. The mutation also results in the loss of ability to produce sclerotia, aerial bacteria, increased filaments and altered mycelial hydrophobicity [26]. In this study, *Sscak1* was observed to exhibit loss of virulence and growth defects, akin to the phenotype observed in cell cycle kinase-associated mutants within the CMGG group. As a result, we hypothesize that *Ss*Cak1 might play a role in fungal growth and cell cycle regulation within *S. sclerotiorum*. However, further verification through subsequent experiments is necessary.

As a cell cycle kinase, CDK-activating kinase (Cak) primarily exerts its activation mechanism through phosphorylation of a conserved threonine residue located in the T-loop region of CDK [27,28]. In the realm of yeasts, *Saccharomyces cerevisiae* harbors a vital *Cak* gene, denoted as *Cak1*, while *Schizosaccharomyces pombe* employs two partially redundant Cak systems, namely the MCS6-MCS2 complex and Csk1, to facilitate the activation of Cdc2 during cellular division [28,29,30]. Intriguingly, within *S. sclerotiorum*, the elimination of *Ss*Cak1 did not yield lethal consequences, as the resulting *SsCak1* knockout mutants exhibited comparable deficiencies in growth and infection levels to the previously reported *SsCDC28* mutants [31]. Consequently, this observation proposes that Cak1 might actively engage in the cell cycle progression of *S. sclerotiorum* by effectuating the phosphorylation of CDC28. Nonetheless, the veracity of this proposition necessitates further validation through subsequent experimental investigations. The association of cell cycle-regulated kinases with appressorium formation has been demonstrated in many important pathogenic fungi [32,33]. In some cases, specific cell cycle phases must be completed prior to attachment formation, as has been described in *M. oryzae*, where completion of the S and M phases is mandatory for attachment formation and function [33]. In other cases, however, the cell cycle must be stopped at a specific cell cycle stage to allow appressorium to form. In *U. maydis*, infectious filaments (where the attachment will differentiate) must stop in the G2 phase [34]. In *M. oryzae*, the lack of cyclin-dependent kinase CDC14 hinders appressorium generation [35]. Considering that the main role of the appressorium is to facilitate the penetration of the fungal hyphae within its host to proliferate after invading plant tissue, the formation of appressorium is subordinate to the regulation of the cell cycle. This affiliation seems reasonable to ensure that normal genetic information is loaded into the invading hyphae during this process. As expected, appressoria were completely absent in *Sscak1* mutants, and when inoculated on wounded *A. thaliana* and *N. benthamiana* leaves (with a dissecting needle), *Ss*Cak1 deletion mutants produced lesion spots, but the area was much smaller than that of the WT strain. Therefore we hypothesize that the loss of *Ss*Cak1 results in disordered cell cycle regulation, thereby affecting the formation of appressorium and leading to the complete loss of pathogenicity in *S. sclerotiorum*. Nevertheless, its function in other aspects of pathogenicity besides facilitating penetration cannot be excluded.

At present, the prevention and control of sclerotinia are almost entirely dependent on chemical fungicides, and the perennial use of medicaments has made sclerotinia produce obvious drug resistance [36]. Thus, the breeding of disease-resistant varieties becomes more and more important for environmentally friendly control strategies. However, strong host monogenic resistance to *S. sclerotiorum* has not yet been found [2]. RNAi-based approach HIGS offers a flexible and environmentally friendly solution for crop protection [37,38,39]. Hence, HIGS technology provides a species-specific and environmentally safe method for the control of *S. sclerotiorum*. Here, *SsCak1* was chosen as the target gene, and the RNAi construct of *SsCak1* was transiently expressed in *N. benthamiana* using HIGS. The results indicated that the silencing of *SsCak1* through HIGS substantially elevated the plants’ resistance to *S. sclerotiorum*. Nevertheless, a limitation of RNAi technology lies in potential off-target effects [40]. However, the expression of *SsCak1* decreased by 60% in the WT strain after infection, which further verified the correctness of HIGS. Therefore, in the genome of *S. sclerotiorum*, genes similar to *SsCak1* that are conserved in pathogens without homology in host plants are expected to be target genes of HIGS.

In summary, a putative *SsCak1* (*sscle_11g085070*) was identified by a combination of forward genetic screening and NGS, and demonstrated that it was persistently expressed during hyphal development and highly expressed during *S. sclerotiorum* infection. A knockout strain of *SsCak1* was subsequently generated by split-tagging and then transformed with a WT copy to generate two complementary strains. Deletion of *SsCak*1 led to defects in mycelium development, sclerotia development, appressorium formation, and penetration, resulting in complete loss of virulence, suggesting that *Ss*Cak1 is essential for both growth and pathogenicity regulation, and thus can serve as a potential target for enhancing crop resistance to *S. sclerotiorum* through HIGS.

## 4. Materials and Methods

### 4.1. Fungal Strains, Plants and Culture Conditions

The WT *S. sclerotiorum* 1980 was cultured and maintained on potato dextrose agar (PDA). The deletion mutant strain and the complementary strain were grown on PDA containing 150 μg/mL hygromycin B (Roche) in a 20 °C culture room, as previously described [41]. WT *A. thaliana* (Col-0), *N. benthamiana* for virulence test were grown in a growth room at 22 °C with a 16 h light/8 h dark cycle.

### 4.2. Inoculation and Virulence Assessment

For inoculation of *S. sclerotiorum* with mycelial suspension, 6 fresh mycelium agar plugs with a diameter of 5 mm were put into a 250 mL flask containing 150 mL PDB and incubated at 22 °C and 150 rpm for 24 h. The resulting mycelial spheres were collected by filtering the medium with filter paper and washed 3 times with ddH_2_O and PDB, respectively. The mycelium balls were ground on ice to homogenize them. The resulting liquid mycelial suspension was then adjusted to OD_600_ = 1.0 with PDB solution, and inoculation was referred to previous standard techniques [42,43]. The expanded detached or un-detached leaves were inoculated with actively growing mycelial agar plugs (1 mm diameter for *A. thaliana* leaves and 6 mm diameter for *N. benthamiana* leaves). Ten leaves each of *A. thaliana* and *N. benthamiana* were inoculated, in each replication. The experiments were replicated at least three times. The inoculated leaves were incubated at 22 °C with 95–100% relative humidity. The lesion size was photographed at 24 hpi.

### 4.3. Screening for Mutants Defective in Virulence

Ascospores were collected from the ascus of the WT strain *S. sclerotiorum* 1980. Ascospores were subjected to UV mutagenesis, as previously described [18]. Screening of mutagenized populations used lettuce leaves and multiple replicates to confirm putative mutants.

### 4.4. Genomic DNA Extraction and NGS

The agar block with a diameter of 5 mm was selected and fresh hyphae were inoculated on PDA covered with cellophane. After 2 days, the mycelium was scraped off with a sterile tip, quick-frozen in liquid nitrogen, and ground into powder. and then genomic DNA was extracted using the cetyltrime-thylammonium bromide method [44]. The crude extract was further purified for NGS with a commercial service (Novogene, Bioinformatics Technology Co., Ltd., Beijing, China). DNA degradation and contamination were evaluated on a 1% agarose gel. The paired-end library was built by Novogene, using Illumina sequencer NovaSeq 6000, and clean reads were aligned to the *S. sclerotiorum* genome (ASM185786v1).

### 4.5. Candidate Genes Identification

The sequence reads from NGS were mapped to the reference genome of WT strain *S. sclerotiorum* 1980. Mutations were identified by SAMtools with default parameters [45]. Annotation of the mutations was performed with germline short variant discovery (SNPs + INDELs) based on GATK best practices [46]. False mutations in repetitive sequences were manually removed as previously described [19].

### 4.6. Target Gene Knockout and Transgene Complementation

Using the genomic DNA of WT *S. sclerotiorum* as a template, the flanking sequences of 1032 bp upstream and 998 bp downstream of *SsCak1* were amplified by PCR and fused with the left and right parts of the hygromycin expression cassette to obtain the split marker fragments by overlapping PCR. The clone primer sequences are shown in Appendix A. The resulting split marker fragments were transformed into WT *S. sclerotiorum* by PEG-mediated protoplast transformation, as previously described [47]. Transformants were selected three times on PDA medium containing 150 mg/L hygromycin. And the transformants were purified by transferring at least three times with a mycelial tip. The deletion of the *SsCak1* gene was verified by amplifying the sequence of *SsCak1* from the cDNA of transformants.

For the genetic complementation of the *SsCak1* deletion mutant, a genomic region containing the full-length fragment of *SsCak1*, including upstream and downstream of the coding sequence, was cloned from WT *S. sclerotiorum* gDNA. This fragment was then cloned into the modified *p*CH-NEO1 vector [48]. Complementary transformants were selected on PDA medium containing 100 μg/mL G418 and then verified by PCR.

### 4.7. RNA Extraction and cDNA Synthesis

Total RNA from *S. sclerotiorum* or *N. benthamiana* leaves with *S. sclerotiorum* inoculated was extracted using the EastepTM Super Total RNA Extraction Kit (Promega, Madison, WI, USA). According to the manufacturer’s instructions, first-strand cDNA was synthesized by the GoScript™ Reverse Transcription System Kit (Promega, Madison, WI, USA).

### 4.8. RT-qPCR

Real-time PCR was performed on a StepOneTM Real-time PCR Instrument Thermal Cycling Block using SYBR^®^ Green Premix Pro Taq HS qPCR Kit II (AG11702, Accurate Biotechnology (Hunan) Co., Ltd., Changsha, China). Use the following PCR program: 40 cycles of 2 min at 94 °C, 15 s at 94 °C and 1 min at 58 °C. The internal reference gene for *S. sclerotiorum* is *Tubulin1*. Relative gene expression levels were analyzed using the 2^−ΔΔCT^ method [49].

### 4.9. Compound Appressoria Observation

*S. sclerotiorum* mycelium plugs of 5 mm were placed on glass slides and cultured for 16 h to observe the formation and number of appressorium. After 16 h of inoculation with *S. sclerotiorum*, onion epidermises were soaked in a 0.5% trypan blue solution for 30 min and then destained using a bleach solution (ethanol: acetic acid:glycerol = 3:1:1). Samples were examined and photographed under an optical microscope (Axio Imager 2, ZEISS, Oberkochen, Germany).

### 4.10. Construction of TRV–HIGS Vectors and Agro-Infiltration in N. benthamiana

Short cDNA fragments of *SsCak1* in *S. sclerotiorum* were amplified by PCR using gene-specific primers with *Eco*RI and *Bam*HI linkers. The resultant PCR product was digested with *Eco*RI and *Bam*HI and cloned into a linearized *p*TRV2 vector. A TRV2-based construct harboring GFP was used as a negative control of VIGS and HIGS [50,51]. A TRV2-based construct harboring phytoene desaturase (PDS) from *N. benthamiana* (*pTRV2:PDS*) was used as a positive control for VIGS efficiency (Senthil-Kumar and Mysore, 2014). Finally, *p*TRV1 and these *p*TRV2-based constructs were separately transformed into *Agrobacterium* tumefaciens GV3101 by electroporation.

The mixed *Agrobacterium* solution was then infiltrated into the first and second leaves of 2-week-old *N. benthamiana* using a needle-free syringe as previously described [36]. *N. benthamiana* plants were then grown in a growth chamber for at least 14 days.

## Figures and Tables

**Figure 1 ijms-24-12610-f001:**
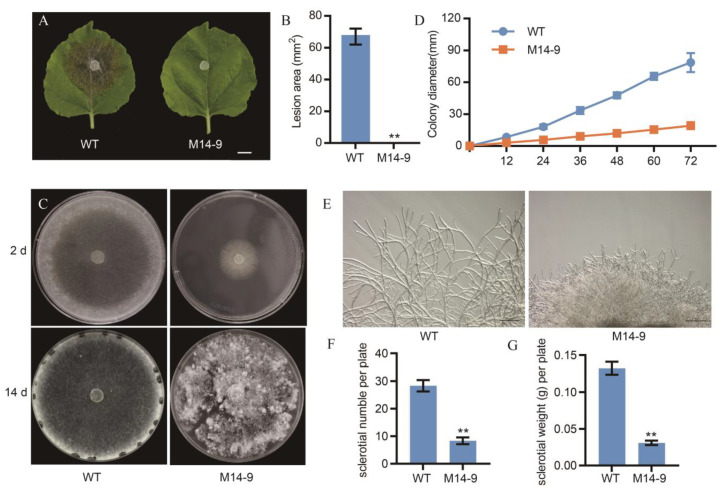
Pathogenicity and mycelial growth phenotypes of M14-9 mutant. (**A**) Necrotic areas caused by wild type (WT) and M14-9 on tobacco leaves at 48 hpi; The experiment was repeated at least three times. Bar = 1 cm. (**B**) Quantification of lesion size of WT and M14-9 on tobacco leaves. Image J was used to quantify the lesion size. ** *p* < 0.01, one-way ANOVA test. (**C**) Mycelial morphology of WT and M14-9 strains after 2 d or 14 d on potato dextrose agar (PDA) media (**D**) Colony diameter of M14-9 and WT strains cultured on PDA plates. (**E**) Morphology of mycelium under light microscope of M14-9 and WT strains. Bar = 200 μm. (**F**,**G**) Sclerotia number (**F**) or weight (**G**) per plate of M14-9 and WT strains. ** *p* < 0.01, one-way ANOVA test.

**Figure 2 ijms-24-12610-f002:**
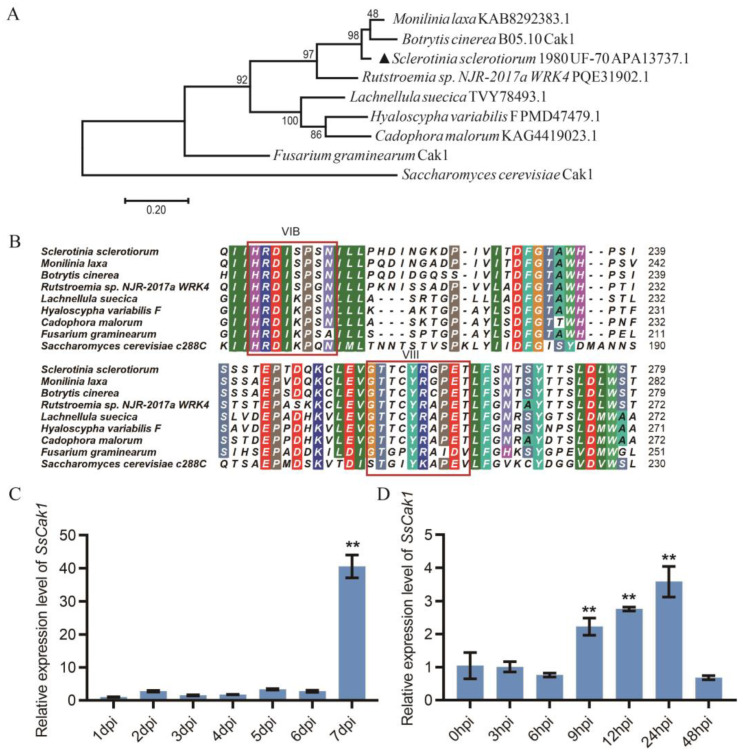
Sequence and expression analysis of the candidate gene in M14-9. (**A**) Phylogenetic analysis of *S. sclerotiorum sscle_11g085070* and other homologous Cak1 from *Botrytis cinerea*, *Monilinia laxa*, *Rutstroemla* sp. *NJR-2017a WRK4*, *Lachnellula suecica*, *Hyaloscypha variabilis F*, *Cadophora malorum*, *Fusarium graminearum* and *Saccharomyces cerevisiae*. Phylogenetic analysis was performed using MEGA 6.0 software with the maximum likelihood method. *Ss*Cak1 was marked with ▲. (**B**) Multiple alignments of *sscle_11g085070* with homologous sequences of *B. cinerea*, *M. laxa*, *R.* sp. *NJR-2017a WRK4*, *L. suecica*, *H. variabilis F*, *C. malorum, F.graminearum and S. cerevisiae*; (**C**) Quantitative real-time reverse transcription-polymerase chain reaction (RT-qPCR) analysis of *SsCak1* expression during different developmental stages of *S. sclerotiorum* grown on potato dextrose agar (PDA) plates. The quantity of *tubulin1* (*Sstub1*) was used to normalize the expression levels of *SsCak1* in different samples. Error bars represent ±SD (*n* = 3). ** *p* < 0.01, one-way ANOVA test. (**D**) Expression analysis of *SsCak1* in *S. sclerotiorum* after being inoculated on *N. benthamiana* leaves. The quantity of *tubulin1* (*Sstub1*) was used to normalize the expression levels of *SsCak1* in different samples. Error bars represent ±SD (*n* = 3). ** *p* < 0.01, one-way ANOVA test.

**Figure 3 ijms-24-12610-f003:**
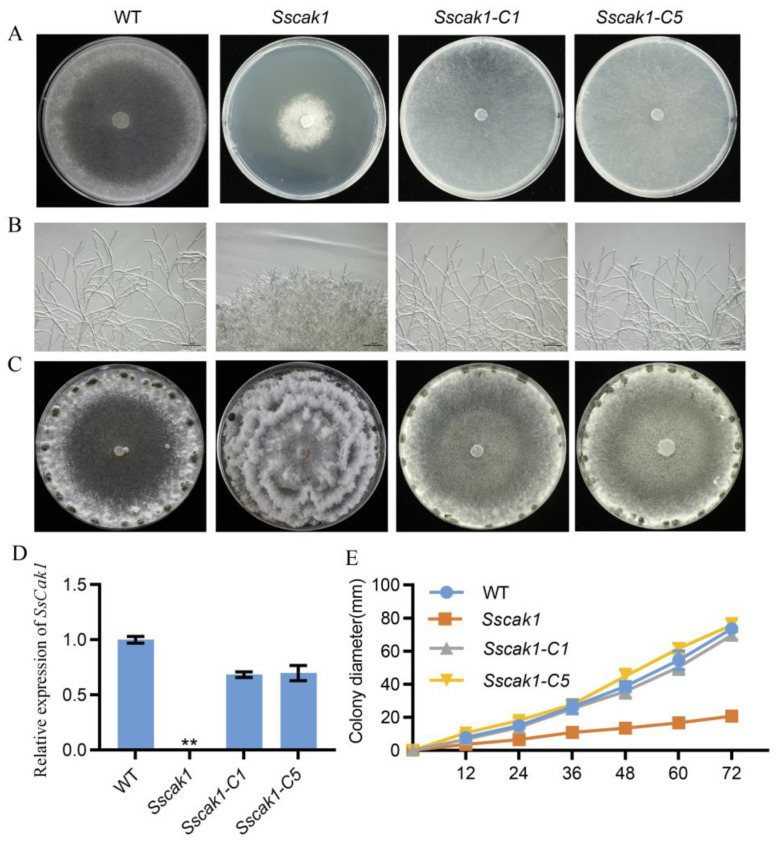
Morphology of the *SsCak1* knockout and complementation strains. (**A**) Colony morphology of WT, *Sscak1*, *Sscak1-C1* and *Sscak1-C5* strains after 2 days on potato dextrose agar (PDA) media. (**B**) Morphology of mycelium under light microscope of WT, *Sscak1*, *Sscak1-C1* and *Sscak1-C5* strains. Bar = 200 μm. (**C**) Sclerotia morphology at 15 days on potato dextrose agar (PDA) media. (**D**) RT-qPCR was used to monitor the expression levels of *SsCak1* in WT, *Sscak1*, *Sscak1-C1* and *Sscak1-C5* strains. The data were normalized to the *Sstub1* transcript level of the WT strain. Error bars represent ±SD (*n* = 3). ** *p* < 0.01, one-way ANOVA test. (**E**) Colony diameter of the WT, *Sscak1*, *Sscak1-C1* and *Sscak1-C5* cultured on PDA plates.

**Figure 4 ijms-24-12610-f004:**
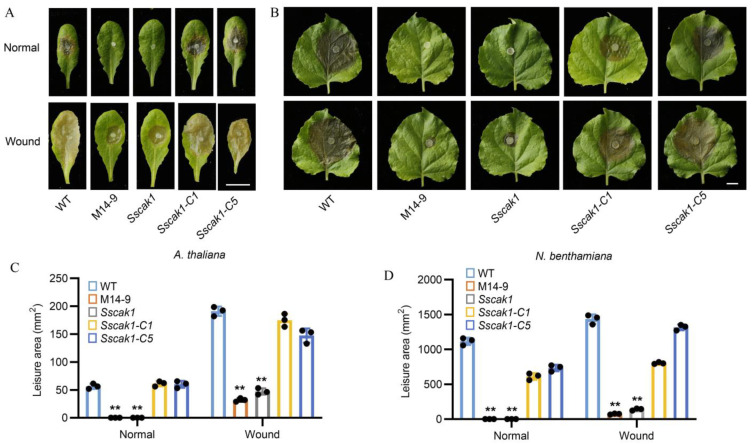
Deletion of *SsCak1* leads to loss of pathogenicity in *S. sclerotiorum*. (**A**,**B**) Lesion area caused by each strain on intact and injured *A. thaliana* (**A**) and *N. benthamiana* (**B**) leaves. The experiment was repeated at least three times. Bar = 1 cm. (**C**,**D**) Statistical analysis of the lesion area in panels above (**C**), *A. thaliana*; (**D**), *N. benthamiana*). Error bars represent ±SD (*n* = 3). ** *p* < 0.01, one-way ANOVA test.

**Figure 5 ijms-24-12610-f005:**
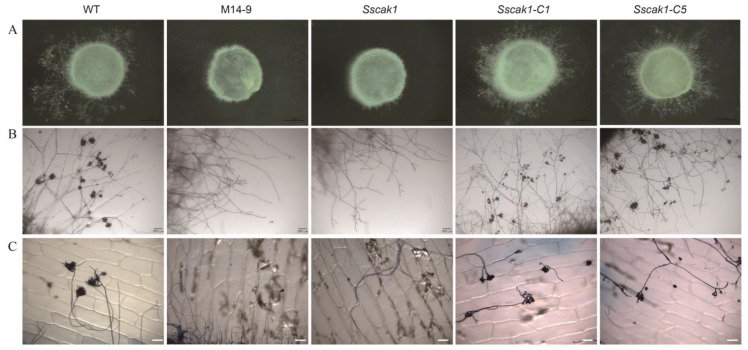
Knockout of *SsCak1* results in defects in appressorium formation and penetration. (**A**,**B**) WT, M14-9, *Sscak1*, *Sscak1-C1* and *Sscak1-C5* strains were placed on glass slides and cultured for 16 h to observe the number of appressoria formed. Bar = 200 μm. (**C**) Penetration assay of WT, M14-9, *Sscak1*, *Sscak1-C1* and *Sscak1-C5* on onion epidermis cells. Invasion hyphae were stained with trypan blue. Bar = 100 μm.

**Figure 6 ijms-24-12610-f006:**
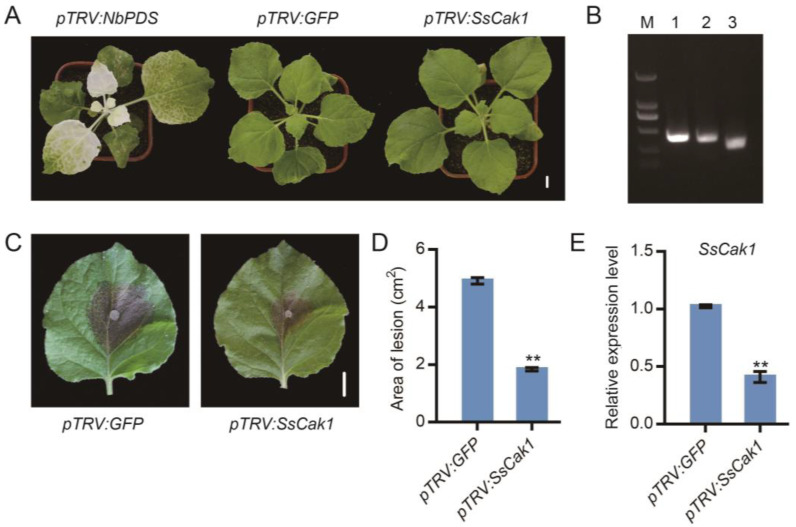
Effects of host-induced gene silencing (HIGS) of *SsCak1* in *N. benthamiana*. (**A**) The phenotype of *pTRV2:SsCak1*-infiltrated plants 14 days after infiltration. Bar = 1 cm. (**B**) Reverse transcription PCR was used to monitor expression levels of target genes in infiltrated plants. M, 2-kb DNA ladder; 1, *pTRV2:GFP*; 2, *pTRV2:PDS*; 3, *pTRV2:SsCak1*. (**C**) Necrotic symptoms on leaves inoculated with WT strain at 2 d post-inoculation (dpi). One of the representative biological replicates is shown. Bar = 1 cm. (**D**) Statistical analysis of the lesion area. Error bars represent ±SD (*n* = 10). ** *p* < 0.01, one-way ANOVA test. (**E**) Gene expression levels of *SsCak1*. The data were normalized to the *Sstub1* transcript levels of the WT strain. Error bars represent ±SD (*n* = 3). ** *p* < 0.01, one-way ANOVA test.

## Data Availability

The data presented in this study are available on request from the corresponding author.

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
