# Peer review of "SsCak1 Regulates Growth and Pathogenicity in Sclerotinia sclerotiorum"

_ijms, 2023, doi:10.3390/ijms241612610_

Round 1

Reviewer 1 Report

Sclerotinia sclerotiorum is a plant fungal pathogen causing stem rot disease in a wide range of important crops. This study by Qin et al. utilized a forward genetic screen in combination with NGS and identified a putative protein kinase SsCak1 as a novel pathogenic factor in S. sclerotiorum. Creating mutant and complemented strains for this gene, the authors demonstrated that SsCak1 is crucial for the pathogenicity of S. sclerotiorum. Using N. benthamiana as a model system, the authors showed that SsCak1 can be a potential target for controlling S. sclerotiorum infection through host-induced gene silencing.

This is a very well-designed and well-executed study. The data presented in the article back all the conclusions very nicely and the findings are very well presented in the manuscript writing.

I have some minor concerns that should be addressed/explained by the authors.

Line 236 – the authors should complete the sentence by adding ‘PCR’ after the word semi-quantitative.

Figure 6B presenting the expression of the target genes quantified by semi-quantitative PCR gel indicates the three different target genes in the host plant targeted by three different constructs. What time point was this semi-quantitative PCR performed? It would help to add the control plant (vector control) and/or other control gene(s) to this data to show that specifically the target genes were downregulated by HIGS.

Author Response

Thank you very much for your thorough review of our manuscript and providing valuable suggestions. We have revised the manuscript according to your comments and suggestions, and provide detailed explanations of how we intend to address them.

Reviewer 2 Report

The manuscript needs some reorganization for the results/discussion sections since several data and added in the discussion part and should be moved. Some sentences needs clarification and must be rewritten. The authors should avoid personalization in the whole manuscript and accurately used Italics only for Latin names and genes, avoiding to use it for fungal strain abbreviations. More revisions and suggestions are enclosed in the partly annotated revision of the manuscript provided.

Some reorganization and improvement are necessary as reported above and in the attached partly annotated manuscript.

Author Response

Thank you very much for your review of our manuscript and providing valuable suggestions. We sincerely appreciate the time and effort you have devoted to this process. We have revised the manuscript according to your comments and suggestions, and provide detailed explanations of how we intend to address them.
